Vepris amaniensis: a morphological, biochemical, and molecular investigation of a species complex

Ciambrone Mary msciambrone@gmail.com 1 2
Langat Moses K. 3
Cheek Martin 2
1 Queen Mary University of London , London , United Kingdom
2 Accelerated Taxonomy, Royal Botanic Gardens at Kew , London , United Kingdom
3 Trait Diversity and Function, Royal Botanic Gardens Kew , London , United Kingdom
Sun Genlou
Electronic publication date: 2024 Sep 25
Publication date: 2024
Volume: 12
Electronic Location ID: e17881
Received 2024 Mar 14; Accepted 2024 Jul 17
Copyright: ©2024 Ciambrone et al.
Copyright year: 2024
Copyright holder: Ciambrone et al.
License: This is an open access article distributed under the terms of the Creative Commons Attribution License, which permits unrestricted use, distribution, reproduction and adaptation in any medium and for any purpose provided that it is properly attributed. For attribution, the original author(s), title, publication source (PeerJ) and either DOI or URL of the article must be cited.
License URL: https://creativecommons.org/licenses/by/4.0/

Keywords: Vepris, Sanger sequencing, Biochemistry, Taxonomy, Vepris usambarensis, Phylogeny, ITS, trnL-F

Funding: The authors received no funding for this work.

==============================
Vepris Comm. ex A. Juss. is a genus of 96 species extending from Africa to India that are distinct in their unarmed stems and their digitately (1-)3(-5) foliolate leaflets, and whose many secondary compounds earn them uses in traditional medicine. Mziray (1992) subsumed six related genera into Vepris, with Vepris amaniensis (Engl.) Mziray becoming somewhat of a dustpan for ambiguous specimens (Cheek & Luke, 2023). This study, using material from the Kew herbarium, sought to pull out novel species from those previously incorrectly filed as Vepris amaniensis, and here describes the new species Vepris usambarensis sp. nov. This species is morphologically distinct from Vepris amaniensis with its canaliculate to winged petioles, 0.5–2.3 cm long inflorescences, 1–3 foliolate leaflets, and hairs on inflorescences and stem apices. Phytochemical analysis attributed seven compounds to Vepris usambarensis: tecleanthine (1), evoxanthine (2), 6-methoxytecleanthine (3), tecleanone (4), 1-(3,4-methylenedioxyphenyl)-1,2,3-propanetriol (5), lupeol (6), and arborinine (7). This is a unique mixture of compounds for a species of Vepris, though all are known to occur in the genus, with the exception of 1-(3,4-methylenedioxyphenyl)-1,2,3-propanetriol (5) which was characterized from a species in the Asteraceae. An attempt at constructing a phylogeny for Vepris using the ITS and trnL-F regions was made, but these two regions could not be used to differentiate at species level and it is suggested that 353 sequencing is used for further research. Originally more than one new species was hypothesized to be within the study group; however, separating an additional species was unsupported by the data produced. Further phylogenetic analysis is recommended to fully elucidate species relationships and identify any cryptic species that may be present within Vepris usambarensis.

Introduction

Vepris Comm. ex A. Juss. is a genus consisting of 96 species (Plants of the World Online, 2023) distributed widely in Africa and Madagascar, with one species on the Arabian peninsula and one in India. Generally evergreen trees and shrubs, they are distinct from other African genera in the Rutaceae due to their digitately (1-)3(5-) foliate leaflets and their unarmed stems. Most species can be found in tropical lowland to submontane forest, with a few found in drier habitats. Vepris species are also used as indicators of healthy, relatively undisturbed forests as they are not known to be pioneers (Cheek et al., 2019).

Like other members of Rutaceae, Vepris species are characterized by gland dots on leaves that are filled with aromatic compounds. Many species are also known to have important secondary metabolites in root and stem tissue (Ombito, Chi & Wansi, 2021). The secondary metabolites in these tissues are utilized all over Africa in traditional medicine (Ombito, Chi & Wansi, 2021). The compounds produced are used in various forms to treat a large number of ailments, from everyday problems such as wounds and sores, to more long lasting issues such as rheumatic pains, infertility, and malaria (Ombito, Chi & Wansi, 2021). A recent review reports that 213 compounds have been isolated from various Vepris species, including alkaloids, quinolones, terpenoids, triterpenoids, flavonoids, and coumarins (Ombito, Chi & Wansi, 2021). Some of these compounds have been tested for bioactivity and have displayed antimicrobial, cytotoxic, anti-protozoal, and insecticidal properties (Mwangi et al., 2010; Langat, 2011; Atangana et al., 2017; Ombito, Chi & Wansi, 2021; Ojuka et al., 2023). These properties make the genus a promising one for pharmaceutical research.

The genus underwent a major taxonomic rearrangement in the 1990s. Mziray (1992) collapsed the genera Araliopsis Engl., Diphasia Pierre, Diphasiopsis Mendonça, Oricia Pierre, Teclea Delile, and Toddaliopsis Engl. into Vepris based on morphological analysis. This reorganization was later confirmed with molecular work done by Morton (2017). However, in Morton’s analysis, species were not well delimited and a better supported and more complete tree would be desirable. In subsuming six genera into Vepris, Mziray transferred the names of 31 species, most of which were from the former genus Teclea. One such species was V. amaniensis (Engl.) Mziray; described (as Teclea amaniensis Engl.) in the Flora of Tropical East Africa as a glabrous shrub with unifoliolate or occasionally 2–3 foliolate leaves, elliptic leaflets with a short broad acumen, numerous gland dots on lower leaflet surfaces, terete or occasionally winged petioles, and glabrous or pubescent inflorescences (Kokwaro, 1982). However, in a taxonomic review of unifoliolate African Vepris, it was found that many of the specimens ascribed to V. amaniensis were disparate from the few that agreed with the protologue of Teclea amaniensis (Cheek & Luke, 2023).Very recently, the description of V. amaniensis has been amended to better match the protologue and so is defined as being completely glabrous, having terete to canaliculate petioles, and always being unifoliolate (Cheek & Luke, 2023). This new delimitation has been utilized here to study the c. 30 specimens, collected from Kenya and Tanzania, that were found to disagree with the Teclea amaniensis protologue.

This study aimed to determine how many distinct taxa reside within this group of specimens. Morphological, biochemical, and molecular methodologies were used to address this question.

Materials and Methods

Morphology

We studied twenty-nine herbarium specimens from the herbarium at the Royal Botanic Gardens, Kew (K) that were previously filed as Vepris amaniensis but were reconsidered here as possibly distinct following an inventory of the available specimens at K by Cheek. All the specimens were collected in the East Usambara, West Usambara, Nguru, and Uluguru Mountains of Tanzania, or from the south eastern coast of Kenya. Measurements of vegetative and floral traits were taken with a ruler or a Leica S6E microscope using a graticule eyepiece measuring to 0.025 mm at maximum magnification. Where appropriate fruit or floral material was available, dissections were performed after rehydration and were photographed under a Leica M165 C dissecting microscope. Measurements of floral parts were taken from these photos using ImageJ (Schneider, Rasband & Eliceiri, 2012). The specimens were sorted into groups based on two distinctive vegetative character states: the absence or presence of winged petioles, and the number of leaflets per leaf. These traits resulted in four groups; winged petiole with unifoliolate leaflets (WU), winged petioles with 1–3 foliate leaflets (WT), canaliculate petioles with unifoliolate leaflets (GU), and canaliculate petioles with 1–3 foliate leaflets (GT). The GU group was then split into two subsections, one with proportionally narrower leaflets (GU lance) and one with proportionally broader leaflets (GU broad), to account for two specimens with distinctively narrower leaflets. Specimens studied, their morphological groupings, biochemical sampling, and GenBank accessions can be found in Table 1.

Table 1 Herbarium samples and GenBank accessions.

All herbarium specimens cited were seen and housed at K. Where samples were pooled to make one biochemical extract they are marked as combined. All specimens were sampled for DNA sequencing.

Morphological
group	RBG Kew
Herbarium
specimen	Biochemistry
sampling,
mass (g)	Sample
name	ITS GenBank
accession	trnL-trnF
GenBank
accession	
GT
(Grooved petioles,
1–3 foliate leaflets)	R.B. Drummond and
J.H. Hemsley 3456					
	Ruffo and
Mmari 2354	0.751	GT(1)	OR470720		
	Luke & al 5242				OR466960	
	Borhidi et al. 85240				OR466963	
	Jon Lovett 263			OR470725	OR466958	
	Andrew R. Marshall 1423	combined
0.8674	GT	OR470727	PP328477	
	R.M. Polhill,
J.M. Lovett 5007	combined
0.8674	GT		PP328476	
	Ruffo and Mmari 2170				OR466965	
GU lanceolate
(grooved petioles,
unifoliolate
lanceolate leaflets)	Ruffo and Mmari 2306	0.7843	GU5	OR470718	OR466968	
	Ruffo and Mmari 1785					
GU broad
(Grooved petioles,
unifoliolate
broad leaflets)	Ruffo and Mmari 2304			OR470717	OR466966	
	Ruffo and Mmari 2243	0.6327	GU4	OR470732		
	P.J. Greenway 4895					
	E.B. Wallace 939				OR466971	
WT
(Winged petioles,
1–3 foliolate leaflets)	Luke and Robertson 1716					
	Luke and Robertson 5848			OR470731	OR466964	
	D. Napper 1380			OR470729		
	Robertson and Luke 4538			OR470730	OR466959	
	R.B. Drummond and
J.H. Hemsley 3802	0.6872	WT(2)		OR466955	
	B.Mohoro UMBCP 329			OR470728	OR466962	
	Kisena 1631	combined
1.8253	WT	OR470724	OR466957	
	S.P. Kibuwa 5459	combined
1.8253	WT	OR470723	OR466956	
	Ruffo and Mmari 1724	combined
1.8253	WT		OR466967	
	Ruffo and Mmari 2307			OR470719	OR466969	
WU
(Winged petioles,
unifoliolate leaflets)	S. Paulo 168					
	S.R. Semsei 1508			OR470721	OR466970	
	Mgaya 157	0.3254	WU(T)			
	Drummond and Hemsley 1101			OR470722		
	Luke and Robertson 1900	0.8317	WU(K)			
	Luke WRQ 18906,
unmounted
specimen from Kenya	6.3125	VAK	OR470726	OR466961	
Samples for DNA only						
Vepris amaniensis	J. Lovett, India Ellis and Alison Keeley 869			OR470733	OR466972	
Vepris sansibarensis	T. Muller 4088			OR470736	OR466976	
Vepris sp. A (from Cabo Delgado)	J.E. Burrows & S.M. Burrows			OR470737	OR466977	
Vepris sp. nov. (from Mabu mt.)	F. Dowsett-Lemaire 2529			OR470735	OR466975	
Vepris trichocarpa	Ian Derbyshire 1072			OR470738	OR466978	
Vepris fischeri (Vepris trichocarpa)	J Timberlake 5948				OR466973	
Vepris hemp	Hemp 7152			OR470734	OR466974	

Mapping of specimens was done with coordinates directly as recorded, or with a combination of locality data and the Index of Collecting Localities for the Flora of Tropical East Africa (Polhill, 1988). Mapping was done with ArcGIS Pro (2021).

The electronic version of this article in Portable Document Format (PDF) will represent a published work according to the International Code of Nomenclature for algae, fungi, and plants (ICN [Shenzhen Code]; Turland et al., 2018), and hence the new names contained in the electronic version are effectively published under that Code from the electronic edition alone. In addition, new names contained in this work which have been issued with identifiers by IPNI will eventually be made available to the Global Names Index. The IPNI LSIDs can be resolved and the associated information viewed through any standard web browser by appending the LSID contained in this publication to the prefix “http://ipni.org/”. The online version of this work is archived and available from the following digital repositories: PeerJ, PubMed Central SCIE, and CLOCKSS

Chemistry

Samples were collected from well preserved and representative herbarium specimens within the morphological groupings, as well as one dried sample sent directly from Kenya (Luke WRQ 18906) which was included due to the large amount of its mass that could be sacrificed for extraction. Material was ground to a powder in a spice grinder and/or a pestle and mortar. The powder was extracted in methylene chloride (CH2Cl2 abbreviated DCM) overnight, vacuum filtered and washed, and then extracted in methanol (MeOH) over the next night. Purification, analysis, and characterization of compounds were done with liquid chromatography, thin plate chromatography (TLC), nuclear magnetic resonance spectrometry (NMR), and high-resolution mass spectrometry (HR-MS).

Separation of compounds for each extraction was done using a 2 cm diameter column packed with silica (40–63 micron Davisil®). Solvent systems and the corresponding compounds eluted are outlined in Table 2. Extracts yielded seven compounds; compound 1 was determined to be tecleanthine (Atangana et al., 2017), compound 2 to be evoxanthine (Ombito, Chi & Wansi, 2021), 3 to be 6-methoxytecleanthine (Atangana et al., 2017), 4 to be tecleanone (Casey & Malhotra, 1975), 5 to be 1-(3,4-methylenedioxyphenyl)-1,2,3-propanetriol (Rahman & Moon, 2007), 6 to be lupeol (Ombito, Chi & Wansi, 2021), and 7 to be arborinine (Langat, Kami & Cheek, 2022).

Table 2 Extraction solvent systems.

Solvent systems and the chemicals found at each stage. Methylene chloride is abbreviated to DCM.

	Solvent system	Compound isolated in fraction	
1.
DCM extracts	Hexane		
	Hexane:DCM 1:1		
	Hexane:DCM 2:80	Lupeol (6)	
	DCM	Tecleanone (4)	
	5% EtOAc in DCM	Arborinine (7), 1-(3,4-Methylenedioxyphenyl) -1,2,3-propanetriol (5)	
	10% EtOAc in DCM		
	20% EtOAc in DCM	Tecleanthine (1)	
	30% EtOAc in DCM	Evoxanthine (2),
6-methoxytecleanthine (3)	
2.
Methanol extracts	DCM		
	5% EtOAc in DCM		
	10% EtOAc in DCM	Tecleanthine (1), evoxanthine (2)	
	1% MeOH in DCM		
	2% MeOH in DCM		
	3% MeOH in DCM		
	10% MeOH in DCM		
	20% MeOH in DCM		

Fractions were tested for purity using aluminum-backed TLC plates (silica gel 60 F254; Sigma-Aldrich, St. Louis, MI, USA). Visualization of the plates was done with UV radiation at 245 nm, an anisaldehyde spray reagent (1% p-anisaldehyde: 2% H2SO4: 97% cold MeOH), and heat. Where fractions were not sufficiently pure a second smaller separation was conducted. 1D and 2D NMR data was recorded from a Bruker 400 MHz Advance NMR instrument at room temperature in either CDCl3 or CD3OD. Chemical shifts (δ) are expressed in ppm with reference solvent peaks in H1 and 13C NMR spectra placed at δH 7.26 ppm and δC 77.23 ppm for CDCL3 and δH 3.31 ppm and δC 49 ppm for CD3OD. Where samples were small and crude extract spectra were near identical within morphological groups, samples were combined to produce a stronger signal for easier chemical characterization (Table 1). The two WU herbarium samples were not pooled despite their small mass due to their distant geographical origins.

Compounds 1, 2, and 3, isolated from Luke WRQ 18906 were dissolved in MeOH and confirmed by mass on a Thermo Scientific Orbitrap fusion mass spectrometer (Thermo Fisher Scientific, Waltham, MA, USA). Compound 5, also isolated from Luke WRQ 18906, could not be confirmed with MS, and the remaining compounds could not be isolated in sufficient quantity or quality to be analyzed.

DNA

The sampling and methods used here are informed by those used in Morton (2017). All samples studied for morphology were sampled for DNA, as well as several other Vepris species for comparison (Table 1). Extractions were carried out with 20–30 mg of leaf material ground in a Mixer Mill until powdered. Samples were incubated for 30 min in a 65 °C isolation buffer solution of CTAB (747 µL) and 2-mercaptoethanol (3 µL). SEVAG solution (750 µL, CHCl3: isoamyl alcohol = 24:1) was added and samples were shaken in an orbital shaker at 250 rev/min for 30 min and then centrifuged at 13,000 rpm for 15 min. The supernatant of each sample was transferred into new tubes with 500 µL isopropanol and stored at −20 °C for three days. The samples were then centrifuged for another 15 min, after which the aqueous phase was decanted, and the pellet washed with 70% ethanol twice, with 15 min of centrifuging between washes. The pellets were allowed to dry overnight at room temperature and then resuspended in 100 µL of water. A 1% agarose gel with ethidium bromide was then loaded for quality check.

Two markers were sequenced, the nuclear transcribed spacer (ITS) region and the plastid trnL-F intron and spacer. For ITS primers 101, 102, 2, and 3 were used. PCR Methodology followed Sun et al. (1994). PCR of the whole region (primers 101–102) was attempted, as well as from the combinations 101-2 and 3-102 to maximize likelihood of successful sequencing. The PCR solution was 25 µL consisting of 1µL sample DNA, water (5 µL), TBT (5 µL), Dream Taq (12.5 µL, Thermo Sci, 4 mM MgCl2), DMSO (2%, 0.5 µL) and 0.5 µL of each primer (0.2 µM). PCR conditions were an initial denaturation and activation for 2 min at 94 °C, then 28 cycles of denaturation for 1 min at 94 °C, annealing for 1 min at 52 °C, and one minute of extension at 72 °C. A final 7-minute extension at 72 °C completed the process.

For trnL-F we used the primers c and f following the method of Taberlet et al. (1991). Once again PCR of the whole region was attempted, with c-d and e-f also performed to maximize likelihood of success. The PCR solution was 25 µL, consisting of 1 µL sample DNA, water (5.5 µL), TBT (5 µL), Dream Taq (12.5 µL, 4 mM MgCl2; Thermo Fisher Scientific), and 0.5 µL of each primer (0.2 µM). PCR conditions were an initial denaturation and activation for 2 min at 94 °C, then 35 cycles of denaturation for 1 min at 94 °C, annealing for 1 min at 50 °C, and two minutes of extension at 72 °C. A final 4-minute extension at 72 °C finished the process.

PCR products were cleaned by using a binding buffer (125 µL, Buffer PB Qiagen, Hilden, Germany) and a PCR clean-up column (NucleoSpin), centrifuging (13,000 rpm, 1 min) the solution through and then washing the column twice with 600 µL of AW1 wash buffer (Qiagen). The columns were transferred to clean tubes and 30 µL of 65 °C EB elution buffer (Qiagen) was added. After 10 min samples were centrifuged at 13,000 rpm for 1 min to draw the DNA through the column.

Cycle sequencing was performed for each successful PCR product. The mixture for ITS primers was 1 µL sequencing buffer, 0.15 µL water, 0.1 µL (2%) DMSO, 0.25 µL BigDye Premix 3.1 (Thermo Fisher Scientific), and 0.5 µL of 1 pmol/µL of corresponding primer to make 2 µL of solution. Depending on DNA concentration, 1–3 µL of PCR product was added, with water added for a final reaction volume of 5 µL. The solution for trnL-F primers is the same as above, except for the exclusion of DMSO. Samples were then all subject to 26 cycles of 10 s at 96 °C, 5 s at 50 °C, and 4 min at 60 °C. Products were then cleaned with NaAC and 100% EtOH, resuspended in water and Sanger sequenced on an Applied Biosystems (Waltham, MA, USA) 3730xl DNA analyzer.

Phylogenetic analysis

Geneious Prime (Geneious Prime, 2023. 1.2) was utilized to combine complementary strands and check base-calling, produce alignments of the obtained sequences using the MUSCLE algorithm, and concatenating the resulting alignments into one matrix. Available ITS and trnL-F sequences for Vepris in GenBank were included in the alignment (Table 3), as well as two species of Zanthoxylum and Fagaropsis angolensis (Engl.) Dale to act as outgroup taxa in subsequent analysis (Morton, 2017). Seven specimens; Ruffo and Mmari 1785, Greenway 4895, Mgaya 157, Luke and Robertson 1900, Paulo 168, Luke and Robertson 1716, and Drummond and Hemsley 3456 failed to be amplified at the PCR stage or produced sequences too noisy to be used and so were not included in the analysis. All other specimens had at least partial sequences of either the ITS or the trnL-F markers and were included.

Table 3 GenBank data used in phylogenetic analysis.

Voucher numbers, references, and accession numbers of DNA data pulled from GenBank.

Samples taken from GenBank	Voucher number	Reference	ITS accession number	trnl-trnF accession number	
Fagaropsis angolensis	529288 CM
529290 CM	Morton (2017)
Morton (2017)	KU193664.1
KU193665.1	KU193632.1
KU193633.1	
Vepris amaniensis	RC70
529281CM	Veldman et al. (2020)
Morton (2017)	MN114145.1
KU193666.1	KU193634.1	
Vepris elliotii	Ranaivojaone 592 (US)	Appelhans & Wen (2020)	MK882482.1	K883751.1	
Vepris eugeneiifolia	5614653 MO	Morton (2017)	KU193684.1	KU193652.1	
Vepris glomerata	5393948 MO	Morton (2017)	KU193686.1	KU193654.1	
Vepris grandifolia	Boyona AF
4244019 MO	Morton (2017)
Morton (2017)	KU193683.1
KU193669.1	KU193651.1
KU193637.1	
Vepris heterophylla	5518017 MO	Morton (2017)	KU193682.1	KU19650.1	
Vepris lanceolata	5769420 MO
Kew 2343	Morton (2017)
Groppo et al. (2008)	KU193685.1	KU193653.1
EU853823.1	
Vepris morogoresis	4959977 MO	Morton (2017)	KU193663.1	KU193631.1	
Vepris nobilis	529291 CM
4598475 MO	Morton (2017)
Morton (2017)	KY508613.1
KU193670.1	KY508614.1
KU193638.1	
Vepris sansibarensis	5750529 MO	Morton (2017)	KU193681.1	KU193649.1	
Vepris simplicifolia	NMK:EA 13545
Chase 1764 K	Mwaura (2020)
Groppo et al. (2008)	MT137500.1	EU853824.1	
Vepris sp.	MP 668	Veldman et al. (2020)	MN257824.1		
Vepris stolzii	5700270 MO
5315965 MO	Morton (2017)
Morton (2017)	KU193680.1
KU193676.1	KU193648.1
KU193644.1	
Zanthoxylum chevalieri	6177214 MO	Morton (2017)	KU193687.1	KU193655.1	
Zanthoxylum deremense	5301622 MO	Morton (2017)	KU193688.1	KU193656.1	

A Bayesian inference analysis was performed with MrBayes (Huelsenbeck & Ronquist, 2001, v3.2.7). The ITS and trnL-F regions were treated as independent partitions in the analysis. The General Time Reversible (GTR) model with a proportion of invariable site and a gamma shape to account for rate heterogeneity among sites (GTR+I+G) was assigned to both partitions. Zanthoxylum chevalieri P.G. Waterman was selected as the outgroup taxon because of its availability as a sister genus with relevant data on GenBank. Posterior probability distribution was estimated using Markov chain Monte Carlo sampling (MCMC) over 7 million generations, sampled every 1000th generation and saving branch length.

Tracer (Rambaut et al., 2018, v1.7.2) was used to determine if all the parameters of the analysis reached a stationary phase. A final consensus tree was compiled in MrBayes (v3.2.7) with a burnin phase of 10% (1,000 trees) and using the option contype = allcompat. The resulting tree was visualized using FigTree v1.4.4 (Rambaut, 2010).

Results and Discussion

Morphology

Vepris amaniensis is delimited as being glabrous, unifoliolate, having terete petioles, and inflorescences 0.9–4(-5)cm long (Cheek & Luke, 2023). The samples measured for this study are morphologically distinct, and are separated from V. amaniensis by their canaliculate to winged petioles, 0.5–2.3 cm long inflorescences, 1–3 foliolate leaflets, and hairs on both inflorescences and stem apices. As such, they are here described as the new species Vepris usambarensis (Figs. 1 and 2).

Figure 1 Vepris usambarensis plate 1.

(A) Habit with top side of petiole at * (B) base of bifoliolate leaf, adaxial, with topside of petiole (C) leaf, adaxial, with profile of midrib (D) leaf, adaxial, with profile of midrib (E) male inflorescence (F) male flower after removal of one sepal and petal (G) female inflorescence (H) female flower after removal of one petal (I) mature fruit, side view (J) mature fruit top side (K) detail of pitting on fruit surface. Material used: A–D, I–K Napper 1380 (K003470010), E, F Marshall et al. 1423 (K003470011), G, H, Lovett 263 (K003470013). Image by Andrew Brown.

Figure 2 Vepris usambarensis plate 2.

(A) Habit (B) adaxial view of trifoliolate leaf and petiole with two variants of petiole top side (C) leaf adaxial with profile of midrib (D) leaf abaxial with profile of midrib (E) male inflorescence (F) inner face of petal showing oil glands (G) stamens and vestigial ovary. Material used: A, Drummond et al. 3802 sheet 3 (K003470007), B–G, Drummond et al. 3802 sheet 2 (K003470008). Image by Andrew Brown.

It was originally hypothesized that there might be more than one new taxon within the specimens studied. However, this study could not generate enough support to clearly separate additional taxa other than the proposed new species V. usambarensis. Samples were originally separated into four groups; those with winged petioles and unifoliolate leaflets (WU), winged and 1–3 foliolate leaflets (WT), canaliculate petioles and unifoliolate leaflets (GU), and canaliculate and 1–3 foliolate leaflets (GT). These two characters, petiole morphology and leaflet number, were the only distinctive characters by which the samples could be divided. All other characters, both vegetative and reproductive, appeared to have too little variation across specimens to be credibly utilized in differentiating groups. The availability of mature flowering structures in this study group was low and so with more complete resources, distinct reproductive characters may become apparent.

The use of leaflet number as a delimiting character proved fairly weak across specimens; many nearly exclusively expressing unifoliolate leaflets only to have one or two multi-foliolate leaflets, or vice versa. Others had a relatively even mix of one, two, and three foliolate leaflets. The plasticity in leaflet number displayed on a single specimen suggests that even on specimens that only included unifoliolate material, it was not possible to rule out a higher number of leaflets elsewhere on the plant from which these specimens were taken. In Vepris, petiole morphology and leaflet number are generally important characters used in delimiting species (Cheek & Luke, 2023), and species are generally able to be defined by a single leaflet and petiole state. That this species has so much variation makes it an interesting one for further taxonomic and phylogenetic study. More reproductive specimens and further DNA sequencing may also prove the presence of more taxa.

This variability of leaflet number poses a question concerning the evolution and ecological relevance of having a single leaflet versus multiple leaflets. That these plants have and express the genes for both could suggest a maximizing of efficiency by using two states with differing benefits, or it could simply be an evolutionary artifact going from one state to another.

With support for leaflet number as a taxon identifier low in this case, demarcating additional taxa solely on petiole morphology could not be justified, and so here the study specimens are all classified together as a new, unusually morphologically variable species separate from V. amaniensis.

Vepris usambarensis Ciam. & Cheek, sp. nov. Holotype: Tanzania, Lushoto district, Mazumbai, West Usambaras, University Forest reserve, 1,480 m, fr, fl, March 1984, Jon Lovett 263 (estimated 4.8°S, 38.5°E, herbarium specimen, K003470013).

Dioecious evergreen shrub 0.5–2.5(-5)m tall, alternate branching, grey bark, internodes 0.45–5.3 cm long, stem diameter at lowest leafy node (1-)1.5–5 mm, puberulent at stem apex, rapidly becoming glabrous within 4 nodes of the stem apex, lenticels sub elliptic, raised, c. 0.3–0.5(−0.8) by 0.2–0.3 mm, coverage up to 40% of surface area after 7 nodes from stem apex, leaflets 1–3 foliolate (where a plant is predominantly 1 or 3 foliolate there will generally be at least one leaflet displaying a higher or lower leaflet number).

Leaflets elliptic, 5.8–17.7(-22.7) cm long, c. (2.2-)2.8–5(-7) cm wide, margin simple, entire, where 2–3 foliolate, lateral leaflets are overall 40–60% more reduced than the median; leaflets acuminate, acumen 0.3–1.9 cm long, 2.5–7 mm wide; secondary veins 13–22 on each side of midrib, brochidodromous, gland dots conspicuous on abaxial side, clear in transmitted light, pale green in reflected light, c. 0.1 mm diameter, density 5–7(-11) dots/mm2.

Petiole (0.4-)0.9–3.6(−5.1) cm long, articulated at top and bottom, base puberulent when it falls within 4 nodes of stem apex, apex and abaxial surface with occasional hairs, hairs simple, thickened pulvinus at apical articulation sometimes present, canaliculate to winged, wings up to c. 2.4(-4) mm wide at apex.

Petiolule 0.1–0.3(−0.5) cm long, inconspicuous, terete, glabrous.

Inflorescences axillary paniculate, 0.5–2.3(−3.8) cm long, all parts densely hairy when young, becoming less dense with age, hairs simple, patent, 0.02–0.07 mm long, yellow upon drying.

Peduncle 0.1–0.3(−0.7) cm long, sparsely hairy to puberulent, hairs simple.

Rachis 0.5–1.1(−1.5) cm long, sparsely hairy to puberulent, hairs simple, internodes 0.1–0.5 cm long, alternate, node bearing 1–3 flowers.

Bracts four lobed, lobes triangular, c. 0.25 x 0.5 mm, cupuliform, resembling sepals, surrounding peduncle, puberulent when young, becoming sparsely hairy with age, hairs simple, not seen intact on mature flowers.

Bracteoles subtending pedicels, as for bracts.

Pedicel 1.3–1.8 mm long, sparsely puberulent, hairs simple, yellow.

Flowers dioecious, male and female both 2.3­-3.3 mm long.

Sepals four, triangular, c. 0.6 mm long x 0.8 mm wide, united from base to c. one third of length, ciliate, occasional gland dots.

Petals four, elliptic, c. 2–2.4 mm long, 0.9–1.39 mm wide, drying golden yellow, thickened tip, petal becomes fully reflexed with age, occasional persistent hairs on outer surface, gland dots present on bud, clustered near apex, drying yellow.

Stamens in male flowers 4(-5), filaments 2–2.7 mm long, dorsiventrally flattened, tapering at top, anthers ovoid to discoid, diameter 0.46–0.59 mm, medifixed. Staminode remnants observed in available female flowers.

Ovary in male flowers vestigial, cone shaped, 0.83–1 mm long, c. 0.34–0.38 mm wide at base, densely covered in semi-appressed simple yellow hairs c. 0.5–0.6 mm long, unilocular, yellow-orange, slightly lobed at bottom around stamens (suspected vestigial disk). In mature female flowers ovary is sub-ovoid, c. two mm long and 1.4 mm wide, unilocular, vesicular, scabrid near base, drying brown, 4–5 orange lobes near base around where staminodes emerge, stigma discoid, c. 1.4 mm diameter, convex, style minute, c. 0.1 mm long.

Fruit a single-seeded berry, ellipsoid to apex slightly beaked, 9–13 mm long x 3–8 mm wide, thinly fleshy, exocarp c. 0.4 mm thick, ripening green, brown purple on drying, 4–5 orange lobes generally persistent on bottom, conspicuous gland dots, yellow to brown, slightly raised, 0.1–0.26 mm diameter, occasional persistent hairs, pedicel accrescent, 2–6 mm long.

Seed tan, ellipsoid, dimensions slightly smaller than in fruit, single longitudinal groove.

Representative specimens examined: (All specimens were seen and housed at K)

Tanzania, Morogoro Rural dist, Mkungwe forest reserve, fr, fl, Aug 13, 2000, B. Mhoro UMBCP 329 (est 6°53′S 37°55′E, K003470017),

Tanzania, Muheza dist, Amani-Kwamkoro road 2miles SE of Amani, fr, July 15, 1953, R.B. Drummond and J.H. Hemsley 3456 (estimated 5°7′S 38°37′E, K003470026),

Tanzania, Muheza dist, Kwamkoro forest reserve, Monga, fr, July 18, 1986, Ruffo and Mmari 2354 (est 5°06′S 38°37′E, K003470022),

Tanzania, Muheza dist, east Usambara mt, Kwamkoro forest trail 3, fl, March 1998, Luke & al. 5242 (est 5°10′S 38°36′E, K003470027),

Tanzania, Muheza dist, fl, fr, June 28, 1987, Ruffo and Mmari 2170 (est 5 °10′S 38°48′E, K003470023),

Tanzania, Muheza dist, Kilanga forest reserve, fr, Aug 24, 1986, Ruffo and Mmari 1785 (est 5°18.5′S 38°38′E, K003470020),

Tanzania, Muheza dist, Kwamkoro forest reserve, fr, July 1, 1985, Ruffo and Mmari 2243 (est 5°10′S 38°36′E, K003470030),

Tanzania, Muheza dist, Mgue Sangerawe, fl, Oct 2, 1937, P.J. Greenway 4895 (est 5°8′S 38°37′E, K003470029),

Tanzania, Muheza dist, Mtai forest reserve, fl, Sep 13, 1996, Kisena 1631 (est c.4°50′S 38°46′E, K003470035),

Tanzania, Muheza dist, Kibwanda to Bulwa foot path, st (leaves), Nov 10, 1981, S.P. Kibuwa 5459 (est 5°3″S 38°41″E, K003470015),

Tanzania, Mvomero dist, Turiani, fr, Nov 1953, S. Paulo 168 (est 6°9′S 37°35′E, K003470038),

Tanzania, Mvomero dist, Mtibwa forest reserve, fr, Nov 1953, S.R. Semsei 1508 (est c. 6°7′S 37°39′E, K003470037),

Tanzania, Mvomero dist, Manyangu forest reserve, fl, July 1957, Mgaya 157 (est c. 6°07′S 37°34′E, K003470033),

Tanzania, Lushoto district, Usambara mts, Mahezangulu forest reserve, fl , Jan 24, 1985, Borhidi et al. 85240 (est 4°56′S 38°31″E, K003470028),

Tanzania, Lushoto dist, Mazumbai, west Usambara, university forest reserve, 4°48′S 38°30′E, fr, fl, Mar 1984, Jon Lovett 263 (K003470013),

Tanzania, Lushoto dist, near Mazumbai HQ, Mazumbai forest reserve, west Usambara Mts, 4°48′S 38°30′E, fl, July 3, 2008, Andrew R. Marshall 1423 (K003470011),

Tanzania, Korogwe dist, Ambangulu tea estate, fr, July 17, 1983, R.M. Polhill, J.C. & J.M. Lovett 5007 (est 5°2′S 38°23′E, K003470024),

Tanzania, Korogwe dist, West Usambaras, Ambangulu estate, st (leaves), Oct 1940, E.B. Wallace 939 (est 5°5′S 38°26′E, K003470031),

Tanzania, Korogwe dist, Lutindi forest reserve, fr, Aug 1989, Ruffo and Mmari 2306 (est c.4°53′S 38°38′E, K003470021),

Tanzania, Korogwe dist, Lutindi forest reserve, fr, Aug 1986, Ruffo and Mmari 2304 (est c.4°53′S 38°38′E, K003470032),

Tanzania, Korogwe dist, Lutindi forest reserve, fl , Aug 24, 1986, Ruffo and Mmari 1724 (est c. 4°53′S 38°38′E, K003470018),

Tanzania, Korogwe dist, fr, Jan 22, 1987, Ruffo and Mmari 2307 (est 5°9′S 38°28′E, K003470016),

Kenya, Kilifi dist, Kombeni reserve valley edge of Kaya Fimboni, fl, Aug 21, 1989, Luke and Robertson 5848 (est 3°54′S 39°36′E, K003470012),

Kenya, Kilifi dist, Pangani Rocks, fl, Aug 16, 1989, Luke and Robertson 1900 (est 3°51′S 39°40′E, K003470036),

Kenya, Kwale dist, Buda Mafisini forest reserve, fr, Feb 24, 1989, Luke and Robertson 1716 (est 4°27′S 39°24′E, K003470014),

Kenya, Kwale dist, Buda Mafisini forest reserve, fr, Nov 3, 1959, D. Napper 1380 (est c.4°27′S 39°24′E, K003470010),

Kenya, Kwale dist, Muhka forest, fr, Feb 19, 1987, Robertson and Luke 4538 (est 4°20′S 39°31′E, K003470019),

Kenya, Kwale dist, Buda Mafisini forest reserve, fl, Aug 16, 1953, R.B. Drummond and J.H. Hemsley 3802 (est 4°27′S 39°24′E, K003470009),

Kenya, Mwele Mdogo forest, Shimba Hills 12 miles SW of Kwale, fr, Feb 4, 1953, Drummond and Hemsley 1101 (est 4°18′S 39°21′E, K003470034).

Distribution: Coastal south-eastern Kenya, East and West Usambara mountains, Nguru mountains, and the Uluguru mountains of Tanzania. (Fig. 3.)

Figure 3 Collection sites of samples.

Specimens were mapped using locality data as recorded and the Index of Collecting Localities by Polhill (1988). Produced with ArcPro. Map data: Ersi, CGIAR, USGS, HERE, Garmin, Fao, NOAA.

Habitat: Restricted to relatively undisturbed habitat, in Tanzania relegating it to submontane to montane evergreen tropical forest. Kenyan specimens are also found in protected areas, though at much lower elevations.

Etymology: Named after the Usambara mountains, where the majority of the specimens were collected.

Phenology: Flowers March to September, and fruits August to February.

Recognition:

Vepris usambarensis can be distinguished from Vepris amaniensis Engl. by its canaliculate to winged petioles, the presence of an indumentum at stem apices and on inflorescences, generally shorter panicles (0.5–2.3(-3.8) cm long), and 1–3 foliolate leaflets. Vepris amaniensis Engl. has terete to canaliculate petioles, glabrous stems and inflorescences, 0.9–4(-5) cm long panicles, and unifoliolate leaflets (Table 4). Representative V. amaniensis specimens, including the neotype: Tanzania, Muheza dist, Amani, fl., May 4, 1922, Salmon G 6171 (K000593352!; isoneotype EA) (Cheek & Luke, 2023) were consulted. Specimen information can be found in the supplemental files. Illustrations of representative V. usambarensis specimens are given in Figs. 1 and 2.

Table 4 Distinguishing characters for V. amaniensis and V. usambarensis.

Character	V. usambarensis	V. amaniensis	
Petiole	Canaliculate to winged	Terete at base to canaliculate at apex	
Indumentum	Puberulent at stem apices and on bracts and sepals	Glabrous	
Inflorescence	0.5–2.3(−3.8) cm long	0.9–4(-5) cm long	
Leaflet number	1–3 foliolate	Always unifoliolate	

Note: Although the neotype of Vepris amaniensis (Engl.) Mziray is stated in Cheek & Luke (2023) to be “Salmon 171”, inspection of the annotated neotype sheet at K (see photo in Supplemental Information) shows that a digit is missing, the correct number is in fact 6171. There is no confusion of the specimen since it is annotated as the neotype by the first author of Cheek & Luke (2023) who is also the 3rd author of the current paper, and other details are the same. In addition, the specimen is (correctly) referred to in the Notes of Vepris amaniensis as “Salmon G 6171”.

Chemistry

Of the specimens that were sampled, seven compounds were found in high enough concentration to be described; tecleanthine (1), evoxanthine (2), 6-methoxytecleanthine (3), tecleanone (4), 1-(3,4-methylenedioxyphenyl)-1,2,3-propanetriol (synonym 3′,4′-methylene ether) (5), lupeol (6), and arborinine (7) (Fig. 4). All compounds described were found in CH2Cl2 extracts, methanol extracts yielded tecleanthine (1) and evoxanthine (2) as well, but any other compounds were too dilute to be discerned.

Figure 4 Extracted and characterized compounds.

Compounds found in DCM extracts. 1. Tecleanthine 2. Evoxanthine 3. 6-methoxytecleanthine 4. Tecleanone 5. 1-(3,4-Methylenedioxyphenyl)-1,2,3-propanetriol 6. Lupeol 7. Arborinine.

As with the attempt to create morphological groupings within the studied specimens, the NMR profiles of the samples were too similar to each other to lend credible support to more than one taxon being present (Figs. 5–6). All samples had tecleanthine (1) as the dominant compound, in some samples such as GT(1) and WU(K) it was nearly the only one present. Evoxanthine (2) was the second most dominant compound, and the rest were notably less dominant (Table 5). The only differentiation between samples were the varying concentrations of compounds, which can be attributed to life stage, time of collection, and local climate, or the absence of very minor ones. Absences could be due to the small sizes of some samples. Minor compounds like lupeol (6) and arborinine (7) were able to be isolated from Luke WRQ 18906 (sample name VAK) and not others likely due to it being over six times the mass of many of the other samples.

Figure 5 NMR spectra of DCM extracts.

H1 NMR spectra of crude CH2Cl2 extracts.

Figure 6 NMR spectra of methanol extracts.

H1 NMR spectra of crude MeOH extracts.

The exact combination of chemicals found in this study are not reported in any other Vepris species studied for biochemistry, though all but one of the compounds identified is known to occur in the genus. Vepris grandifolia (Engl.) Mziray and Vepris trichocarpa (Engl.) Mziray, both species that occur in tropical east Africa, are reported with similar mixtures including tecleanthine, tecleanone, evoxanthine, lupeol, arborinine, and 6-methoxytecleanthine (Ombito, Chi & Wansi, 2021). The only compound not described before from a Vepris species is 1-(3,4-methylenedioxyphenyl)-1,2,3-propanetriol (5), classed as a lignan. It was first described from extracts taken from the roots of Dendranthema zawadskii var. latilobum (Maxim) Kitam. (synonymized with Chrysanthemum naktongense Nakai), a temperate member of the Asteraceae native to north east Asia and used medicinally in Korea (Rahman & Moon, 2007). It is not known from any other species. That these two species from disparate clades, continents, and climates produce the same chemical seems unlikely, and while its presence was confirmed with NMR analysis, it could not be confirmed with mass spectrometry analysis. Further investigation of the presence of this lignan in this species is recommended.

Lignans and alkaloids have been recognized for their potent pharmacological potential, and are known to contain compounds important to medicine (Saleem et al., 2005; Barker, 2019; Cui et al., 2020). Tecleanthine (1), evoxanthine (2), 6-methoxytecleanthine (3), lupeol (6), arborinine (7), and 1-(3,4-methylenedioxyphenyl)-1,2,3-propanetriol (5) are known to have bioactive properties including antioxidant, antiprotozoal, cytotoxic, antimicrobial, antiplasmodial, and antifeedant properties (Popp & Chakraborty, 1964; Lwande et al., 1983; Muriithi et al., 2002; Rahman & Moon, 2007; Mwangi et al., 2010; Dongfack et al., 2012; Nouga et al., 2016; Atangana et al., 2017). Tecleanone (4) has not been studied for any bioactivity, but it is considered an intermediary in the chemical pathway to make both tecleanthine (1) and evoxanthine (2) (Dagne et al., 1988; Singh & Bharate, 2006).

Table 5 Chemical components of extracted samples.

Chemical compounds found in each sample. Compounds are considered major when they are readily apparent on the NMR spectrogram and are comparable in height to the reference peak. Compounds that do not meet this criteria are considered minor.

Sample	Tecleanthine	Evoxanthine	6-methoxy
tecleanthine	Tecleanone	1-(3,4- Methylenedioxyphenyl) -1,2,3-propanetriol	Lupeol	Arborinine	
GT	Major	Major		Minor	Minor		Minor	
GU4	Major	Major		Minor	Minor			
GU5	Major	Major	Minor	Minor	Minor			
GT(1)	Major	Minor		Minor	Minor			
VAK	Major	Major	Minor	Traces	Minor	Minor	Minor	
WT	Major	Major	Minor	Minor	Traces		Traces	
WT(2)	Major	Major	Minor	Minor				
WU(T)	Major	Major	Minor		Minor		Minor	
WU(K)	Major	Traces		Traces	Minor	Minor	Traces	
	

Phylogeny

The use of the ITS and trnL-F regions here was modeled on the approach of Morton (2017), as this study is the most complete molecular work done at species level on Vepris to date. In Morton’s study, it is stated that discerning species with these regions is difficult, a conclusion supported here. Utilizing these two regions, isolated from each specimen studied, as well as those included from other Vepris species on GenBank, our consensus tree did not have any node with posterior probability over 0.26 (Fig. 7). There is a loose grouping of the specimens that were collected from Kenya, as well as a loose separation of the winged specimens from the canaliculate specimens. However, with the very small support for these relationships, these conclusions cannot be supported.

Figure 7 Phylogeny of both samples and GenBank data.

Phylogeny produced with MrBayes. The symbols indicate the sample’s morphological group; red stars indicate GT, blue triangles indicate WT, orange hexagons indicate WU, pink crosses indicate GU4, and green crosses indicate GU5. The letter preceding the symbols indicate as follows: k for Kenya, u for the Usambara Mts, and m for the Morogoro region of Tanzania.

It appears that these two regions are sufficiently invariable across Vepris and so should not be relied upon alone in future phylogenetic analyses. Future studies in this group would require approaches that provide a greater amount of information, such as RADseq and targeted enrichment (e.g., Angiosperm353; Johnson et al., 2019) to produce a better supported phylogeny and reveal any cryptic species that Vepris may contain.

Conclusion

The new species Vepris usambarensis is here described as being distinct from V. amaniensis, the name under which its specimens had previously been filed. This species can be confirmed to produce tecleanthine (1), evoxanthine (2), 6-methoxytecleanthine (3), tecleanone (4), 1-(3,4-methylenedioxyphenyl)-1,2,3-propanetriol (5), lupeol (6), and arborinine (7). These are all chemicals known to be found in Vepris, save for the lignan which is known from a Chrysanthemum native to east Asia. The known bioactivity of both the lignan and the alkaloids present, like in many Vepris, give it pharmacological potential.

It was originally hypothesized that there would be more than one taxon within the study group due to the high levels of morphological variation observed in characters usually of value in differentiating species in Vepris. However, morphologically and biochemically any further delineations could not be supported. Molecular work was undertaken to try to add additional support; however, the chosen regions, ITS and trnL-F, failed to produce a tree at the species level with enough support. These regions were chosen following Morton (2017), and we conclude that in Vepris, they have little to no value for differentiating species.

Given the variation of morphology observed, it is recommended that further phylogenetic research be conducted with other sequencing approaches. A more complete tree of Vepris than the one produced by Morton (2017) with more support is desired and further investigation of the phylogenetic diversification of species versus morphology in the genus would shine light on the evolution and ecology of the genus as a whole.

Supplemental Information

Supplemental Information 1 Supplimentary table and NMR spectra

Additional table for compounds detailing in which fraction they were found. Additional H1 NMR spectra of isolated compounds.

Supplemental Information 2 DNA data for the trnL-F region of the samples studied

Aligned sequences, formatted as for Genbank.

Supplemental Information 3 DNA data for unaligned trnL-F region of two samples

Unaligned samples. Formatted as for Genbank.

Supplemental Information 4 ITS genbank accession OR470717-21

Supplemental Information 5 ITS genbank submission OR470722-38

Supplemental Information 6 Sample morphological data

Supplemental Information 7 V. amaniensis neotype

Image of neotype published by Cheek 2023 for Vepris amaniensis, Borhidi et al. 85340.

Thank you to Yanisa Olaranont and Dr. Eduard Mas-Claret for help in the lab and instrument training. Thank you to Lazlo Csiba for training and assistance with molecular work. Thank you to Quentin Luke for providing the substantial specimen from Kenya, and thank you to Dr. Felix Forest for advising on phylogeny building.

Additional Information and Declarations

Competing Interests

Author Contributions

Data Availability

New Species Registration

The authors declare there are no competing interests.

Mary Ciambrone conceived and designed the experiments, performed the experiments, analyzed the data, prepared figures and/or tables, authored or reviewed drafts of the article, and approved the final draft.

Moses K. Langat conceived and designed the experiments, performed the experiments, analyzed the data, authored or reviewed drafts of the article, and approved the final draft.

Martin Cheek conceived and designed the experiments, analyzed the data, authored or reviewed drafts of the article, and approved the final draft.

The following information was supplied regarding data availability:

The DNA sequences are available at GenBank: OR466955–OR466978, OR470717–OR470738, and PP328476–PP328477.

Measurements of the herbarium specimens that are included in the article.

The following information was supplied regarding the registration of a newly described species:

Vepris usambarensis urn:lsid:ipni.org:names:77345528-1

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
