# Peer review of "Vepris amaniensis: a morphological, biochemical, and molecular investigation of a species complex"

_PeerJ, doi:10.7717/peerj.17881_

## Round 0.1 · original submission · Minor Revisions

· Academic Editor

Minor Revisions

We have received the comments from two reviewers. Please revise your manuscript accordingly.

·

Basic reporting

no comment

Experimental design

no comment

Validity of the findings

no comment

Additional comments

Vepris amaniensis: A morphological, biochemical, and molecular investigation of a species complex
Review
Roy E. Gereau, Missouri Botanical Garden

The manuscript presents an analysis of the Vepris amaniensis species complex from a morphological, biochemical, and molecular perspective and concludes with the publication of one new species of the genus Vepris. Here I have concerned myself only with the morphological and taxonomic aspects of the work and the formal presentation of the new species, leaving the biochemical and molecular aspects to the expertise of others. I have, however, reviewed the entire manuscript for language and clarity of presentation and have made tracked changes and comments throughout on the attached Word document.

In general, the taxonomic content is clearly presented and justifies the two principal conclusions: 1) that a number of specimens previously included in Vepris amaniensis do not correspond to the original concept of that species; and 2) that those specimens, despite their variability in several usually diagnostic characters, are most properly considered as a single new species, Vepris usambarensis. Tracked changes on pages 1-6 of the manuscript concern only minor corrections that need no further comment. Beginning with the new species description on page 7, however, some more substantive issues arise:

1. On page 7, line 248, the authors must add the abbreviation “sp. nov.” or equivalent to indicate their intention to publish a new species.
2. Vepris usambarensis is described as a new species with J. Lovett 263 at K as its holotype. It would be highly desirable to cite the barcode number of this specimen if it has one.
3. The holotype of Teclea amaniensis, Warnecke in Herb. Amani 516K, was destroyed at Berlin. A neotype, Salmon 171, has been designated at K, with an isoneotype at EA. A direct comparison of the holotype of V. usambarensis and the neotype of Vepris/Teclea amaniensis is essential to fix the application of the two names. This could logically be placed in one of two places: 1) immediately following the holotype designation on page 7; or 2) Under “Recognition” on pages 9-10. Regardless of placement, this really needs to be done.
4. The statements about the presence or absence of stamens/staminodes in the descriptions of the female flower on lines 284, 285, and 290 seem to contradict each other and must be resolved.
5. The section “Representative specimens examined” is very confused in content and presentation.
a. Specimens are listed in seemingly random order, and it would be highly desirable to group the cited specimens together by country and political district.
b. District names for Tanzanian specimens are irregularly indicated – sometimes by the old designations used in the Flora of Tropical East Africa, sometimes by their modern equivalents, and sometimes not at all. I have corrected these to the best of my ability and am confident that they are all accurate now.
c. A few post facto estimated geographic coordinates for older specimens are given to an excessive degree of precision, e.g. with seconds of latitude or longitude given to three or four decimal places. It is not possible to be that precise with the kind of data available, and these should be rounded to the nearest whole minute of latitude or longitude.
d. Two specimens are cited with only a District name and estimated coordinates, but no definite locality. These specimen labels should be checked for the locality details.
6. The summary of geographical distribution is somewhat confused and misleading. Based on the cited specimens, all Tanzanian collections came from four mountain ranges: East Usambara, West Usambara, Nguru, and Uluguru. I have edited the text accordingly.

These are all of my substantive comments; remaining tracked changes and comments are relatively self-explanatory. When these concerns have been satisfactorily addressed, the article will represent an important advance in our understanding of these plants in East Africa.

·

Basic reporting

The manuscript overall is well written. The authors were successful in clearly establishing their hypotheses based on existing literature, as recent as the Cheek & Luke 2023 publication. This provided sufficient context and relevance as to why the authors chose to purse this study.
The authors provided sufficient raw data at the morphological, biochemical, and molecular levels to support their inferences.

Experimental design

The research question is well defined to begin with. The choice of using the presence or absence of winged petioles, and the number of leaflets per petiole made up for a meaningful groups of samples to study the hypothesis. The use of morphological, biochemical, and molecular data provided layers of evidence to substantiate the observations in the study.
A great detail of the methods used was provided, especially the molecular section (PCR conditions, etc.). However, the analytical/biochemistry extraction protocols are vaguely described. It was just the Line 113, that states "Extractions were then done in methylene chloride and then methanol." Additional details in Table 2 failed to provide a great deal of clarity, in case another researcher aims to replicate the study for a different objective. It goes without saying it adds to the length of the paper and more work for the authors, which is understandable.

Validity of the findings

Chemistry Findings: Despite the additional detail was provided in Table 2, it still gives an impression that a powder of the sample is mixed with a solvent of interest and used for analytical/biochemical analyses. The absence of subsequent purification steps may partly explain the absence of some compounds that the authors attributed to small size of samples [Lines 354-363]. This also explains the presence of compound 5 with NMR but not mass spectroscopy [Lines 374-376].

Line 360- Considering the authors emphasis on how the local climate affects their observations, it would be interesting to know their perspective on the difference in samples collected at their original habitat vs the samples that they obtained from the herbarium at the Royal Botanic Gardens. How different is the climate, soil conditions, etc between the two places?

Additional comments

Line 369- The sudden introduction of ligan 1-(3,4-methylenedioxyphenyl)-1,2,3-propanetriol is confusing despite multiple readings. It would be beneficial to introduce compound 5 belonging to the class of lignans, if that is the case. It gives an impression that compound 5 and lignan compound 5 are different,

Lines 109-112: Despite the sample Luke WRQ 18906 being the most abundant, and procured originally from Kenya (the most native compared to the samples from the Botanical Garden), the results from the sample are not presented in good detail. It provides the reader with a good reference, if those results are available.

---

## Round 0.2 · Minor Revisions

· Academic Editor

Minor Revisions

Please find appended, the comments from reviewer 1 who suggests some further minor revisions

·

Basic reporting

no comment

Experimental design

no comment

Validity of the findings

no comment

Additional comments

The first author has satisfactorily incorporated most of my previous suggested changes into the revised manuscript and explained the points questioned in my comments. I have marked a rather small number of remaining suggested changes on the attached manuscript file.

The only remaining serious issue to be addressed is the citation of the neotype of Teclea amaniensis Engl. and its comparison with the holotype of the new species Vepris usambarensis. In the revised text, the neotype is cited as R. Salumon 6171 at K (barcode K000593352). In the first author's response to reviewers, a neotype is cited as Salumon 1617 with the same barcode; and another neotype is cited as Salumon 6142 (K000593351). This is clearly impossible, because a neotype can consist of only a single specimen in a single herbarium, not two specimens from different gatherings. On the Tropicos database, the neotype is shown as Salumon 171 at K, with an isoneotype at EA, but I do not have access to the source of this information. The authors must verify that the neotypification has in fact been effectively published, not just annotated on one or more specimens at Kew. If a single neotype has already been designated, then it should be more explicitly compared with the holotype of V. usambarensis than in the current text. If not, then the neotypification must be performed in this paper so that the types of the two names can be compared and the identities of the names established. I would suggest that the third author of this manuscript is in an ideal position to do the needed work.

·

Basic reporting

No comment

Experimental design

No comment

Validity of the findings

No comment

Additional comments

The revised manuscript addressed the reviewer comments in sufficient detail. And the authors deserve a word of appreciation in this regard.

---

## Round 0.3 · accepted · Accept

· Academic Editor

Accept

The revised version can be accepted for publication